# Spatial Ensemble: a Novel Model Smoothing Mechanism for Student-Teacher Framework

**Tengteng Huang    Yifan Sun    Xun Wang    Haotian Yao**[*]    **Chi Zhang**

**Megvii Technology**
{huangtengteng, yaohaotian, zhangchi}@megvii.com,
sunyf15@tsinghua.org.cn, bnuwangxun@gmail.com

## Abstract

Model smoothing is of central importance for obtaining a reliable teacher model in the student-teacher framework, where the teacher generates surrogate supervision signals to train the student. A popular model smoothing method is the Temporal Moving Average (TMA), which continuously averages the teacher parameters with the up-to-date student parameters. In this paper, we propose "Spatial Ensemble", a novel model smoothing mechanism in parallel with TMA. Spatial Ensemble randomly picks up a small fragment of the student model to directly replace the corresponding fragment of the teacher model. Consequentially, it stitches different fragments of historical student models into a unity, yielding the "Spatial Ensemble" effect. Spatial Ensemble obtains comparable student-teacher learning performance by itself and demonstrates valuable complementarity with temporal moving average. Their integration, named Spatial-Temporal Smoothing, brings general (sometimes significant) improvement to the student-teacher learning framework on a variety of state-of-the-art methods. For example, based on the self-supervised method BYOL, it yields $+0.9\%$ top-1 accuracy improvement on ImageNet, while based on the semi-supervised approach FixMatch, it increases the top-1 accuracy by around $+6\%$ on CIFAR-10 when only few training labels are available. Codes and models are available at: `https://github.com/tengteng95/Spatial_Ensemble`.

## 1   Introduction

Recent years have witnessed the success of the student-teacher framework in self-supervised [1–5] and semi-supervised [6–8] learning tasks. In this framework, the teacher is responsible for generating surrogate supervision signals to guide the learning of the student on unlabeled data, for example, category IDs for classification, feature patterns for contrastive learning, *etc.*

Many important works under this framework share a common model smoothing technique, *i.e.*, the Temporal Moving Average (TMA), to generate the teacher from the student model. For example, Mean Teacher [6] for semi-supervision averages historical student weights as the teacher. MoCo [1] for self-supervision employs a slowly progressing encoder (teacher), driven by a momentum update with the query encoder (student). Another state-of-the-art method for self-supervision BYOL [2] employs a similar strategy to update the target (teacher) network with a slow-moving average of the online (student) network. We thus recognize that model smoothing with TMA is of central importance for obtaining a reliable teacher for these works.

In this paper, we rethink the mechanism of TMA and summarize its importance as two-fold. First, TMA enables the teacher to absorb parameters from different student editions and thus benefits from

---

[*]corresponding author

35th Conference on Neural Information Processing Systems (NeurIPS 2021).

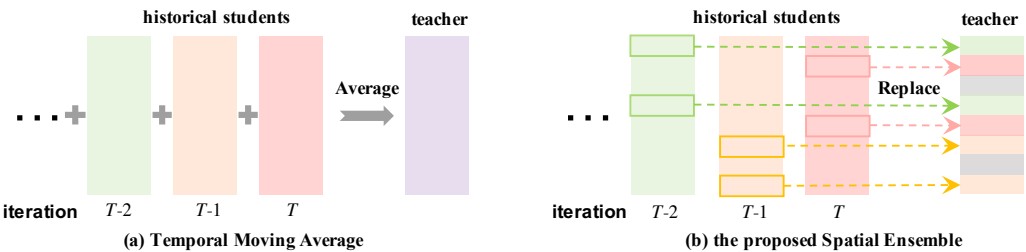

Figure 1: Illustration of Temporal Moving Average and Spatial Ensemble.

the model ensemble. Second, TMA constrains the variance of the teachers to be small to avoid inconsistent labels produced during two adjacent updates. An abrupt change to the pseudo labels will cripple training and possibly leads to optimization divergence. This is particularly important for self-supervised learning where annotations are unavailable.

We propose a novel model smoothing mechanism named "Spatial Ensemble". Compared with TMA, Spatial Ensemble facilitates similar model smoothing benefits with a different mechanism, as illustrated in Figure 1. In TMA, the teacher absorbs each historical student as a whole. A temporally near student is absorbed into the teacher with a larger weighting factor due to the Exponential Moving Average effect. In contrast, in Spatial Ensemble, the teacher randomly picks up one or more small fragments of the student to replace the corresponding fragments of the teacher during each update. All the fragments are weighted equally in the teacher model. In this way, Spatial Ensemble stitches different fragments of previous student models into a unity, yielding the "Spatial Ensemble" effect. In spite of their fundamental differences, Spatial Ensemble facilitates similar benefits of model ensemble and smoothed update. The smoothed update is maintained by replacing only a small portion of parameters at each iteration. On the challenging self-supervised learning task, which is very sensitive to the quality of surrogate supervision signals, we observe that Spatial Ensemble achieves comparable performance with temporal moving average. It shows that our Spatial Ensemble can serve as a basic model smoothing technique for the student-teacher framework.

An important advantage of Spatial Ensemble lays in its complementarity to Temporal Moving Average. To explore their complementary benefits, we integrate both the temporal smoothing and spatial smoothing mechanism into a unified method named Spatial-Temporal Smoothing (STS). STS does not directly absorb a whole student, or simply replace a fragment. Instead, it updates a fragment of the teacher model with a temporal moving average of the corresponding fragment of the student. This "mix-up" model smoothing benefits from the mutual complementarity of temporal moving average and Spatial Ensemble. Experiments on both self-supervised and semi-supervised tasks show that using STS for model smoothing obtains general (sometimes significant) improvement over a majority of state-of-the-art methods.

We summarize our contribution as follows:
1. We propose a novel model smoothing mechanism named Spatial Ensemble. Compared with the popular temporal moving average, Spatial Ensemble achieves a comparable model smoothing effect with a different mechanism.

2. We integrate the spatial and temporal smoothing mechanism into a unified method named Spatial-Temporal Smoothing. STS benefits from the mutual complementarity of two mechanisms and consistently improves student-teacher methods on self- and semi-supervised learning tasks.

3. We notice two recognizable advantages of the model learned through STS, including strong robustness against various data corruptions, and superior transferring capability to the object detection task.

## 2   Related Work

In this section, we present a brief review of the development on semi- and self-supervised learning, as well as the student-teacher framework. We mainly focus on the recent deep learning literature related to our method.

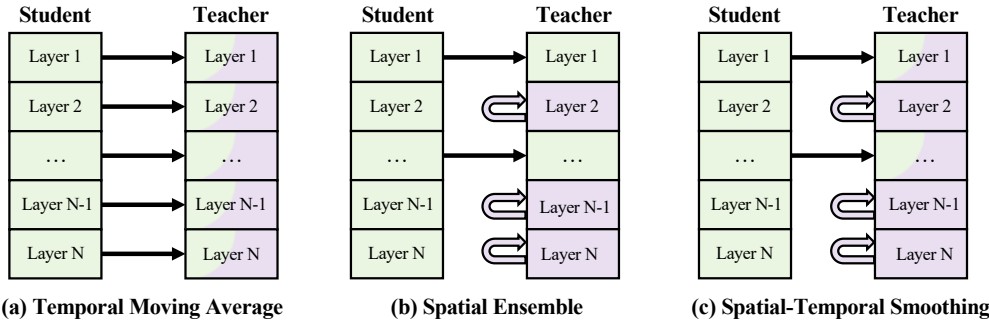

Figure 2: Illustration of three different model smoothing approaches TMA, SE, and STS.

**Semi-supervised learning** aims at achieving comparable performance to fully supervised methods, usually with a small portion of labeled data and the rest as unlabeled data. It is of great importance for real scenarios where annotations are difficult/expensive to acquire, such as medical data. Pseudo labels [9, 10] and consistency regularization [6, 7, 11] are two essential techniques in many recent semi-supervised approaches. Pseudo-Labeling [9] generates soft pseudo labels for the unlabeled data using the logits predicted by the model from the latest epoch. Laine *et al.* [10] improve the pseudo label generation process by leveraging a temporal ensembling prediction of different epochs. Mean-teacher [6] further simplifies this process by introducing a teacher model, which is a temporal ensembling of the student model through the exponential moving average, to generate pseudo labels directly. Besides, mean-teacher employs a consistency regularization to encourage similar predictions from the teacher and the student model. FixMatch [7] assumes that the consistency should also hold between images applied with augmentations of different degrees, and applies both a weak and a strong set of data augmentations.

**Self-supervised learning** largely narrows down its performance gap with supervised approaches and is receiving increasing attention in the community. Recent self-supervised methods can be roughly categorized into two genres, *i.e.*, approaches based on pretext task and approaches based on contrastive learning. There are many kinds of insightful pretext task designs. One common method is generating surrogate supervision signals by feature clustering [12–14], image augmentation (usually rotation [15]), relative order of clipped image patches [16–18], *etc.* Another mainstream approach is to design a generative pretext task, *e.g.*, image colorization [19, 20], image inpainting [21] and image denoising [22], and usually employ auto-encoders or generative adversarial networks [23] to transfer images. Instead of designing a pretext task, contrastive learning approaches [1, 4, 24–29] aim at maximizing the feature similarities with augmented views produced from the identical image sample, while minimizing those from different image samples.

**Student-teacher framework** is a popular architecture that has been extensively used in many recent state-of-the-art methods, especially for semi- and self-supervised learning tasks. Usually, temporal moving average is employed to generate the teacher from historical student models. The optimization goal is to ensure the prediction consistency between the student and the teacher. Besides pulling different views of the same image closer, MoCo [1] employs a memory bank to store negative samples and pushes views from different images apart. BYOL [2] removes negative samples and leverages an asymmetric student-teacher architecture to prevent collapsing. An extra predictor is appended to the online (student) network to regress the output of the target (teacher) network.

## 3   Method

**Preliminaries.** Given a same network architecture, $\Theta^{\mathcal{T}}$ and $\Theta^{\mathcal{S}}$ are the parameters of the teacher and student model. Both $\Theta^{\mathcal{T}}$ and $\Theta^{\mathcal{S}}$ contains $n$ units of parameters, *i.e.*, $\Theta^{\mathcal{T}} = \{\theta_1^{\mathcal{T}}, \theta_2^{\mathcal{T}}, \cdots, \theta_n^{\mathcal{T}}\}$ and $\Theta^{\mathcal{S}} = \{\theta_1^{\mathcal{S}}, \theta_2^{\mathcal{S}}, \cdots, \theta_n^{\mathcal{S}}\}$. A unit may correspond to a layer, a neuron or a channel, depending on the desired spatial granularity. To analyze the model update in consecutive iterations, we further define them as functions of the iteration $t$, *i.e.*, $\Theta^{\mathcal{T}}(t) = \{\theta_i^{\mathcal{T}}(t)\}$ and $\Theta^{\mathcal{S}}(t) = \{\theta_i^{\mathcal{S}}(t)\}$.

## 3.1 Spatial Ensemble

We present a new model smoothing method named Spatial Ensemble (SE), as illustrated in Figure 2 (b). During each iteration, SE randomly picks up some units of the student model to replace the corresponding units of the teacher model, leaving the remaining parts of the teacher unchanged. We formulate this process as:

$$\Theta^{\mathcal{T}}(t) = \{\mathcal{P}_i \theta_i^{\mathcal{T}}(t-1) + (1 - \mathcal{P}_i)\theta_i^{\mathcal{S}}(t-1)\}, \tag{1}$$

where $\mathcal{P}_i \sim Ber(p)$ (*i.e.*, the Bernoulli distribution with success probability $p$) is a binary variable. We note that all $\mathcal{P}_i \sim Ber(p)(i = 1, 2, \cdots, n)$ are mutually independent to each other and have the same success probability $p$. It determines whether the $i$-th unit of the teacher parameters is to be preserved. Specifically, if $\mathcal{P}_i = 1$, the $i$-th unit parameters is to be preserved. If $\mathcal{P}_i = 0$, the $i$-th unit is to be replaced. Since $\mathcal{P}_i \sim Ber(p)$, larger $p$ results in lower replacing frequency and higher proportion of preserved units. Therefore, we name $p$ as the "preserving probability".

For clarity, we omit the iteration indicator in all the equations in the following part of this paper and rewrite Equation 1 as:

$$\Theta^{\mathcal{T}} \Longleftarrow \{\mathcal{P}_i \theta_i^{\mathcal{T}} + (1 - \mathcal{P}_i)\theta_i^{\mathcal{S}}\}. \tag{2}$$

To investigate the mechanism of SE, we quantitatively analyze the smoothing effect on both the model parameters and the generated surrogate supervision signals. Specifically, we calculate the Mean-Square Errors (MSE) between teachers at two adjacent epochs, as well as the MSE between their outputs. We visualize the MSE values at the early training stages (epoch 1 to 20) in Fig. 3, from which we make two observations:

**Remark 1: Spatial Ensemble smooths the teacher update.** It is observed that without model smoothing, the MSE of both model parameters and supervision signals maintain relatively high values, indicating strong vibration during the teacher update. In another word, the teacher model and its output signals undergo substantial instability. In contrast, when using the TMA and the proposed SE to update the teacher, the MSE of both model parameters and supervision signals are much smaller (than Non-smoothing), yielding a smoothed teacher update.

**Remark 2: Spatial Ensemble promotes convergence.** Using TMA and SE, the MSE values gradually decrease as the training epochs increases. Such result indicates that the up-to-date teacher model gradually become closer to its prior edition, therefore approaching convergence. Without TMA or SE, the Non-smoothing mode maintains a roughly constant MSE value. We thus infer that Spatial Ensemble (as well as the TMA) promotes convergence of student-teacher framework.

We provide the pseudocode of SE in Algorithm 1, which is easy to implement with only a few lines of code. In Section 4.3, we further analyze the effect of $p$ on student-teacher learning performance.

---

**Algorithm 1** Pseudocode of SE.

```
# net_s, net_t: student/teacher network
# m: momentum for TMA
# p: probability of a student layer
    belonging to the unpicked fragment.

params_s = net_s.parameters()
params_t = net_t.parameters()
for s, t in zip(params_s, params_t):
if random.rand() < p:
   continue
else:
   t = s
```

**Algorithm 2** Pseudo code of STS.

```
# net_s, net_t: student/teacher network
# m: momentum for TMA
# p: probability of a student layer
    belonging to the unpicked fragment.

params_s = net_s.parameters()
params_t = net_t.parameters()
for s, t in zip(params_s, params_t):
if random.rand() < p:
   continue
else:
   t = m * t + (1-m) * s
```

---

## 3.2 Spatial-Temporal Smoothing

We further integrate TMA and SE into a unified model smoothing method named Spatial-Temporal Smoothing (STS), to benefit from their mutual complementarity. STS picks up several units in

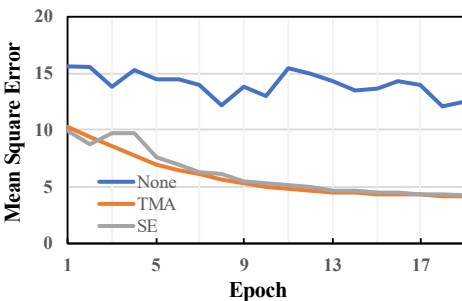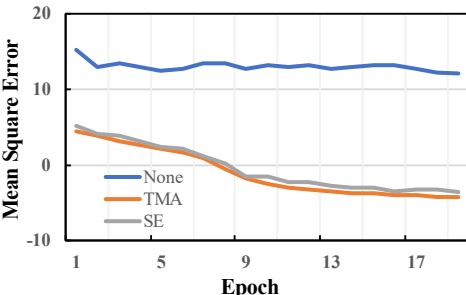

Figure 3: Illustration of mean square error of teacher parameters (**Left**) and generated supervision signals (**Right**) at two adjacent epochs. **None** represents no model smoothing is used. **TMA** adopts a momentum of 0.999. $p$ is set to 0.999 for **SE**. The vertical coordinates are arranged in Log scales.

the student model and averages them with the corresponding units in the teacher model, which is formulated as:

$$\Theta^{\mathcal{T}} \Longleftarrow \{\mathcal{P}_i \theta_i^{\mathcal{T}} + (1 - \mathcal{P}_i) m \theta_i^{\mathcal{T}} + (1 - \mathcal{P}_i)(1 - m) \theta_i^{\mathcal{S}}\}, \tag{3}$$

where $m$ is the momentum for TMA.

STS degenerates to SE or TMA, given specification of hyper-parameters. When $m = 0$, STS degenerates into SE. Similarly, when we set the preserving probability $p = 1$, which results in $\mathcal{P}_i \equiv 0$, STS degenerates into TMA:

$$\Theta^{\mathcal{T}} \Longleftarrow m\Theta^{\mathcal{T}} + (1 - m)\Theta^{\mathcal{S}}, \tag{4}$$

Algorithm 2 illustrates the pseudocode of STS. Similar to SE, STS is easy to implement and can be integrated into most teacher-student frameworks. In Section 4.3, we present detailed analysis of how $p$ and $m$ influence the performance of STS.

# 4 Experiments

## 4.1 Baselines and Implementation Details

We evaluate the effectiveness of the proposed Spatial-Temporal Smoothing by applying it to the state-of-the-art self-supervised approaches (MoCo [1] and BYOL [2]) and semi-supervised method (Fix-Match [7]). We use the official implementation of MoCo[2] and re-implement BYOL and FixMatch using Pytorch [30]. All experiments are conducted on a machine with 8 RTX2080 GPUs. The masking probability $p$ is set to 0.7/0.5/0.5 for BYOL/MoCo/FixMatch, respectively. We dump the teacher model at the end of training. The self-supervised models are evaluated by following the common protocol of linear classification [1, 2, 4, 31] while semi-supervised performance are assessed with top-1 accuracy metric. We carefully follow both the training and evaluation settings of [1, 2, 7] in their original paper for fair comparison. More details are given in the following.

**MoCo** [1] consists of a query model and a key model, which simulates the same role as the student and the teacher model, respectively. We adopt MoCo v2 with ResNet-50 as our baseline and use a memory bank of size 65536. The initial learning rate is set to 0.03 and adjusted by cosine learning rate scheduler [32]. Following the original paper, we train the model using SGD with momentum of 0.9, weight decay of 0.0001, and a mini-batchsize of 256. Both the query and key model adopt the same data augmentations as SimCLR [4], *i.e.*, random crop, horizontal flip, color jitter, Gaussian blur, and gray-scale conversion.

**BYOL** [2] reformulates the self-supervised learning task as a regression task which maximizes the similarity of feature representations from the teacher and the student model. We train BYOL with Synchronized batch normalization [33] and use a mini-batchsize of 256. The basic learning rate is 0.1 and is decayed by cosine strategy [32]. The first 10 epochs are warmed up with a factor of 0.001 and

[2]https://github.com/facebookresearch/moco

gradually increased to the basic learning rate. We adopt SGD with momentum of 0.9 as the optimizer and set weight decay to 1e-4. BYOL uses the same set of data augmentations as SimCLR [4] as listed above, and adopts solarization with probability of 0.2 for the teacher model.

**FixMatch** [7] uses identical student and teacher network in the training phase while employing the TMA teacher for evaluation. We use Wide ResNet-28-2 [34] and Wide ResNet-28-8 as the backbone networks for CIFAR-10 and CIFAR-100, respectively. We use SGD with momentum 0.9 as the optimizer and train the model for $2^{20}$ iterations following the original paper. Each mini-batch comprises of 64 labeled images and 448 unlabeled images (*i.e.*, 1:7). The initial learning rate is 0.03 and is decayed by cosine scheduler [32]. The weight decay is set to 0.0005 and 0.001 for CIFAR-10 and CIFAR-100, respectively. For data augmentation, the student only uses random crop and random horizontal flip. Besides these two augmentations, the teacher additionally adopts random augmentation [35].

**Evaluation.** Linear Evaluation is a common metric for measuring whether the feature representation learned by the self-supervised model is good. Typically, a classifier is appended to the feature encoder and finetuned on the down-streaming task. During the whole process, the parameters of the feature encoder, as well as the BN statistics, are fixed. We train the classifier appended to MoCo using the SGD optimizer for 100 epochs. The initial learning rate is 30 and is decayed at the 60th and 80th epoch with factor 0.1. For BYOL, the basic learning rate is 0.2 and is adjusted according to the cosine strategy during the full 80 epochs.

## 4.2 The Effect of SE and STS

We first evaluate the effect of SE on the self-supervised task. We select MoCo and BYOL as the baseline models and train them for 200 epochs. As shown in Table 1, disabling the model smoothing mechanism in MoCo and BYOL leads to failure of convergence, which is also observed in their original papers. The experimental results indicate the importance of model smoothing in these self-supervised approaches. When combined with SE, MoCo/BYOL achieves 66.7%/67.8% top-1 accuracy, demonstrating the effectiveness of SE in model smoothing.

We further validate the effectiveness of STS which integrates both the temporal and spatial smoothing strategies. As is shown in Table 2, compared to typical TMA, our method boosts the performance of MoCo (BYOL) from 67.5% (71.8%) to 67.8% (72.7%) in terms of top-1 accuracy, indicating the valuable complementarity of the spatial and temporal smoothing mechanism.

| Method | Smoothing | top-1 | top-5 |
|--------|-----------|-------|-------|
| MoCo   | -         | fail  | fail  |
| MoCo   | SE        | 66.7  | 87.0  |
| BYOL   | -         | fail  | fail  |
| BYOL   | SE        | 67.8  | 88.3  |

Table 1: Analysis of the effectiveness of SE as a model smoothing approach.

| Method | Smoothing | top-1 | top-5 |
|--------|-----------|-------|-------|
| MoCo   | TMA       | 67.5  | 88.0  |
| MoCo   | STS       | 67.8  | 88.1  |
| BYOL   | TMA       | 71.8  | 90.6  |
| BYOL   | STS       | 72.7  | 90.9  |

Table 2: Analysis of the superiority of STS over TMA.

## 4.3 Ablation Studies

In this section, we evaluate the effect of hyperparameters in STS and then investigate Spatial Ensemble of different granularities. BYOL is adopted as the default network for the following experiments. All the ablation experiments are conducted on the ImageNet dataset [36] and trained for 200 epochs unless noted otherwise.

**Effect of hyperparameters $p$ and $m$.** To explore how $p$ and $m$ influences the student-teacher learning performance, we conduct extensive experiments by selecting $m$ from {0, 0.9, 0.99, 0.999} and varying $p$ 0.1 to 0.9 with an interval of 0.2. We first investigate the effect of $p$ by setting $m = 0$. Under this case, STS degenerates into SE. As is shown in Figure 4, BYOL fails to converge when $p = 0$ while achieves increasing performance with the increase of $p$, demonstrating the importance of model smoothing. Figure 4 illustrates the results with different combinations of $p$ and $m$. It can be observed that STS yields better performance than typical TMA ($p = 0$) in most cases. We also notice an interesting fact that with the increase (decrease) of $m$, the model achieves its best performance

at relatively smaller (larger) value of $p$. For example, $p = 0.99$ is the best choice when $m$ is set to 0.9. However, for $m = 0.99$, a smaller $p$ of 0.7 yields much better performance than larger $p$ of 0.99. These results demonstrate the complementarity of spatial and temporal smoothing, as well as the important role of model smoothing for self-supervised learning based on the student-teacher framework.

**Granularity of Spatial Ensemble.** SE may have various granularities, corresponding to a layer, a channel, or a neuron as the smallest unit. When integrating SE into STS, we set the granularity to layer-wise by default. In this section, we investigate the impacts of different granularities, *i.e.*, layer-wise (`LW`), channel-wise (`CW`) and neuron-wise (`NW`). Specifically, for a student parameter with shape $C_{in} \times C_{out}$, where $C_{in}$ and $C_{out}$ denote the number of input and output channels, respectively. For `CW`, we randomly pick up a small set of indexes along with the $C_{out}$ dimension and replace the corresponding teacher parameters. Similarly for `NW`, we randomly select neurons from the whole student model and replace the corresponding neurons of the teacher model. As shown in the top of Table 4, SE using `NW` outperforms its counterpart using `LW` and `CW` and achieves comparable performance with TMA. The result reveals that SE itself benefits from the fine-grained spatial ensemble. Besides, three variants of STS result in comparable performance, indicating the complementarity of temporal smoothing and spatial ensemble. When combined with temporal smoothing, `LW` is capable to produce a representative teacher model as good as its fine-grained counterpart `NW`. Considering that the time cost of `LW` is lower than both `CW` and `NW`, yet their performances are close, we use `LW` as default.

### 4.4 Self-supervised evaluation of STS

In this part, we substitute the model smoothing mechanism in BYOL [2] with our STS and compare the resulted model to other state-of-the-art methods on self-supervised learning. Furthermore, we investigate the robustness of our method to corrupted data. Finally, we evaluate the transfer capability of our method on the object detection task.

**Comparison with State-of-the-art method.** Table 3 presents the linear evaluation results of our method compared with the state-of-the-art self-supervised method on ImageNet [36]. Using BYOL as the base model, STS yields competitive top-1 accuracies of 73.0% and 74.5% under 300 and 1000 epochs, outperforming previous state-of-the-art approaches. It is worth noting that our model is trained with a smaller batch size of 256, while BYOL requires 4096. According to the results in the original BYOL paper, when the batch size decreases from 4096 to 256, its top-1 accuracy drops around than 0.6%.

**Robust to corruptions on ImageNet-C.** We further evaluate the robustness of our method to data corruptions. Specifically, we select ImageNet-C [37], which is a standard benchmark dataset for evaluating the model robustness to common corruptions and perturbations, as our testbed. There are 15 different corruption types in ImageNet-C. Besides, each type has 5 corruption intensities. In the left part of Figure 5, we give some image examples of corruptions from ImageNet-C and the corresponding top-1 accuracy improvements of STS over TMA. Combining the right part of Figure 5, we observe that STS outperforms TMA under all the five corruption intensity settings and yields an improvement of around 1.5% in terms of the mean top-1 accuracy of 75 corruption combinations. Besides, considering a single corruption from these 75 combinations, STS leads to an improvement of around $0.5\% \sim 5.6\%$. These results demonstrate the superiority of our method in terms of the robustness to corruptions and further reveal the advantage of STS in enhancing the representative capability of the teacher model.

**Transfer learning in object detection.** We further evaluate the generalization ability of the learned representation using our approach for the more challenging object detection task. In Table 5, we compare our method with state-of-the-art self-supervised approaches. All the self-supervised models are trained for 200 epochs on the ImageNet dataset and then finetuned with the joint split of VOC2007 trainval and VOC2012 train. And their results are taken from SimSiam [38]. For fair comparison, we carefully follow the training and evaluation settings in SimSiam [38]. In more detail, we use the same detection codebase[3] as SimSiam [38] and MoCo [1]. The default detector is Faster-RCNN [39] with a backbone of R50-C4. We finetune all the layers and BN statistics for 24,000 iterations with 16 images per batch. The image scales is set to [480, 800] pixels in the traning stage while 800

---

[3]https://github.com/facebookresearch/moco/tree/master/detection

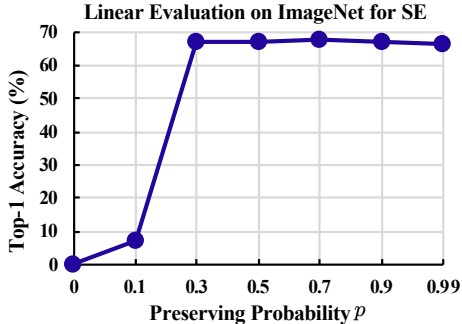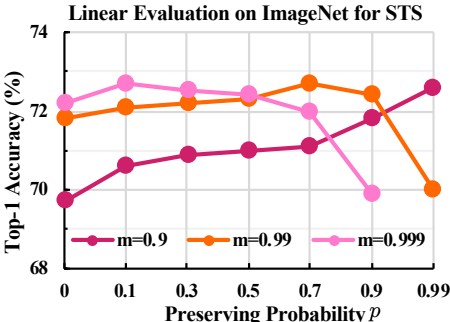

Figure 4: Linear evaluation results of SE (*Left*) and STS (*Right*) under different hyperparameter settings.

for inference. The initial learning rate is 0.02 and is decreased by a factor of 0.1 and 0.01 at when reaching 18,000 and 22,000 iterations. We can observe from Table 5 that the representation learned by our approach can be well transferred to downstream detection task and achieves comparable or superior performance than previous approaches, especially on the $AP_{75}$ metric.

| Method | Epoch | top-1 | top-5 |
|---|---|---|---|
| Jigsaw [40] | 90 | 45.7 | - |
| InstDis [41] | 200 | 56.5 | - |
| BigBiGAN [42] | - | 56.5 | - |
| Local Agg. [43] | - | 60.2 | - |
| CPC v2 [25] | 200 | 63.8 | 85.3 |
| SimCLR [4] | 200 | 66.6 | - |
| MoCo v2 [1] | 200 | 67.5 | - |
| PCL [44] | 200 | 67.6 | - |
| InfoMin Aug [24] | 200 | 70.1 | 89.4 |
| Swav [3] | 400 | 70.7 | - |
| SimSiam [38] | 400 | 70.8 | - |
| BYOL [2] | 300 | 72.5 | 90.8 |
| **Ours** (BYOL + STS) | 300 | **73.0** | **91.3** |
| PIRL [45] | 800 | 63.6 | - |
| SimCLR [4] | 1000 | 69.3 | 89.0 |
| InfoMin Aug [24] | 800 | 73.0 | 91.1 |
| MoCo v2 [1] | 800 | 71.1 | - |
| SimSiam [38] | 800 | 71.3 | - |
| Swav [3] | 800 | 71.8 | - |
| BYOL [2] | 1000 | 74.3 | 91.6 |
| **Ours** (BYOL + STS) | 1000 | **74.5** | **91.8** |

Table 3: Comparison with state-of-the-art self-supervised methods on ImageNet under linear evaluation. All the experiments are based on ResNet-50 and pretrained with two $224 \times 224$ views without multi-crop augmentation.

| Method | Smoothing | top-1 | time |
|---|---|---|---|
| TMA | - | 71.8 | $1.0\times$ |
| SE | LW | 67.8 | $0.97\times$ |
| SE | CW | 70.8 | $1.02\times$ |
| SE | NW | 71.6 | $1.80\times$ |
| STS | LW | 72.7 | $0.99\times$ |
| STS | CW | 72.8 | $1.04\times$ |
| STS | NW | 72.8 | $1.82\times$ |

Table 4: Comparison of SE and STS with different granularities.

| pretrain | VOC07+12 detection | | |
|---|---|---|---|
| | $AP_{50}$ | AP | $AP_{75}$ |
| supervised | 81.3 | 53.5 | 58.8 |
| SimCLR | 81.8 | 55.5 | 58.8 |
| MoCo v2 | 82.3 | 57.0 | 63.3 |
| SwAV | 81.5 | 55.4 | 61.4 |
| SimSiam | 82.4 | 57.0 | 63.7 |
| BYOL | 81.4 | 55.3 | 61.1 |
| **Ours** | 82.3 | 57.2 | 64.4 |

Table 5: Transfer learning on VOC object detection benmark.

## 4.5 Semi-supervised Evaluation of STS

This part is to evaluate the performance of our STS when integrated with FixMatch [7] for semi-supervised learning. Following the common settings [6, 7], we use CIFAR-10 and CiFAR-100 as the benchmark datasets.

**CIFAR-10.** We evaluate our method on CIFAR-10 under four settings, *i.e.*, using 2, 4, 25, 400 labels per class. As shown in Table 6, STS achieves consistently superior performance than TMA under these four different settings, yielding an improvement of 6.63%, 1.83%, 0.21%, and 0.17%,

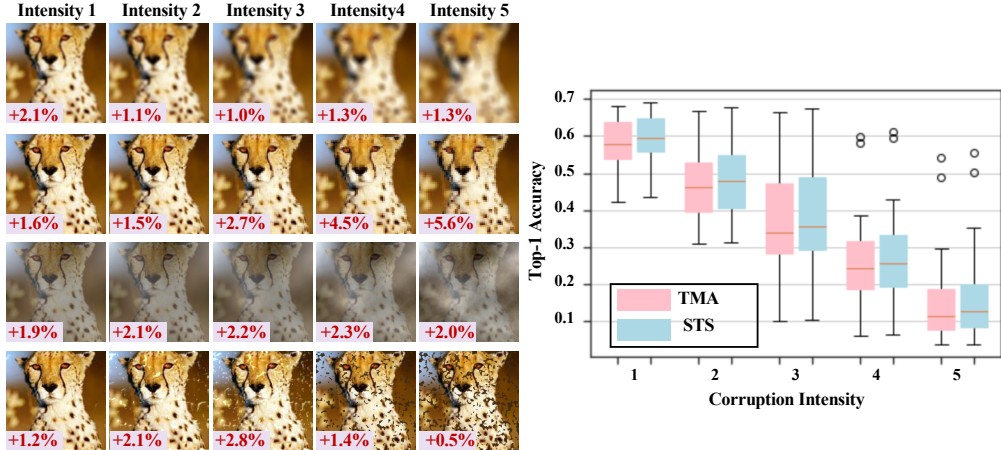

Figure 5: *Left*: Examples of 4 corruption types, namely *defocus blur*, *pixelate*, *fog*, and *spatter*. The numbers attached to the left-bottom corner denote the improvement of STS over TMA on the corresponding corruption type and intensity in terms of top-1 accuracy. *Right*: Comparison of top-1 accuracy between TMA and STS on ImageNet-C.

| Method | CIFAR-10 | | | | CIFAR-100 | | | |
|---|---|---|---|---|---|---|---|---|
| | 20 labels | 40 labels | 250 labels | 4000 labels | 200 labels | 400 labels | 2500 labels | 10000 labels |
| ∏-Model [46] | - | - | 45.74 | 85.99 | - | - | 42.75 | 62.12 |
| Pseudo-Labeling [9] | - | - | 50.22 | 83.91 | - | - | 42.62 | 63.79 |
| Mean Teacher [6] | - | - | 67.68 | 90.81 | - | - | 46.09 | 64.17 |
| MixMatch [47] | - | 52.46 | 88.95 | 93.58 | - | 32.39 | 60.06 | 71.69 |
| UDA [48] | - | 70.95 | 81.18 | 95.12 | - | 40.72 | 66.87 | 75.50 |
| ReMixMatch [49] | - | 80.90 | 94.56 | 95.28 | - | 55.72 | 72.57 | 76.97 |
| FixMatch | 86.68 | 92.06 | 94.89 | 95.77 | 42.78 | 52.54 | 70.75 | 76.53 |
| FixMatch (STS) | 93.31 | 93.89 | 95.10 | 95.94 | 45.85 | 55.25 | 72.22 | 77.92 |

Table 6: Comparison with state-of-the-art semi-supervised methods on the CIFAR-10 and CIFAR-100 benchmark datasets under the top-1 accuracy metric.

respectively. Although the numerical improvement is relatively small when 250/4000 labels are available, it is actually non-trivial since the results have already been very close to the performance upper bound (97.01%), which leverages all labels of 50,000 CIFAR-10 images. The comparison with 20/40 labels is more valuable since how to narrow the performance gap between semi-supervised and supervised learning with limited annotations is a heated research topic in the community. The remarkable improvement achieved by STS demonstrate its superiority and indicates the potential practical value in real scenarios where usually few labeled data is available.

**CIFAR-100.** In Table 6, we further provide the results on CIFAR-100 benchmark dataset under four settings, *i.e.*, using 2, 4, 25, and 100 labels per class. Similar to the phenomenon observed in CIFAR-10, STS achieves noticeable improvements when only a few labels (200/400 labels) are given, outperforming the baseline by 3.07% and 2.71%. Besides, with more labels (2500/10000 labels), STS also achieves consistent improvements of 1.47% and 1.39%, demonstrating the effectiveness of our approach.

## 5 Conclusion

This paper reveals a new mechanism for model smoothing from the spatial ensemble perspective. In addition to the capacity of model smoothing, an important advantage of spatial ensemble is its complementarity to the popular temporal moving average. To benefit from such complementarity, we integrate temporal moving average and spatial ensemble into a unified model smoothing method named Spatial-Temporal Smoothing (STS). Extensive experiments on self-supervised and semi-supervised tasks show that STS brings general improvement to many state-of-the-art methods. Besides, STS achieves competitive results on the ImageNet-C dataset and the downstream object detection task, demonstrating its good generalization ability.

## Acknowledgment

This paper is supported by the National Key R&D Plan of the Ministry of Science and Technology (Project No. 2020AAA0104400).

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
