# OpenReview forum: "Spatial Ensemble: a Novel Model Smoothing Mechanism for Student-Teacher Framework"
_NeurIPS.cc/2021/Conference — NeurIPS 2021 Poster_

### Official Review · Reviewer_ytVw · 2021-07-15

**Rating:** 6
**Confidence:** 3

**Summary:**

Different from the popular EMA mechanism in self-supervised and semi-supervised learning, this paper proposes a new mechanism for model smoothing in spatial ensemble perspective as complementary to the popular temporal moving average.  By combining the proposed Spatial Ensemble and the popular Temporal Ensemble, this work proposes to integrate temporal moving average and spatial ensemble into a unified model smoothing method, named as Spatial-Temporal Smoothing (STS). Experiments on self-supervised and semi-supervised tasks show that STS brings improvement to some methods.

**Limitations And Societal Impact:**

1. Based on the results in Table1 and Table2, the Spatial Ensemble (SE) works worse than the TMA, and it can only work when cooperating with TMA, so basically, STS can only be viewed as an upgraded version of TMA.
2. Comparing the default layer-wise (LW) with channel-wise (CW), it seems that CW achieves much better performance in Table 4, also slightly better performance than LW in STS with nearly the same time. Why does this work choose LW as the default method rather than CW.
3. The improvements of STS on ImageNet are incremental in most methods.

**Main Review:**

This work rethinks current Temporal Moving Average (TMA) in the student-teacher framework, and the proposed Spatial-Temporal Smoothing (STS) is novel for the student-teacher framework, which also achieves superior results on multiple baseline methods in self-supervised and semi-supervised learning.

**Time Spent Reviewing:**

3

---

> ### Author Response · Authors · 2021-08-10
> **Thanks and response to concerns**
>
> We thank the reviewer for the constructive comments.
>
> > 1. Using SE alone seems to achieve much lower accuracy than TMA (Table 1 and Table 2), so STS can be viewed as an upgraded version of TMA.
>
> We beg to differ. We think that neither TMA nor SE dominates STS. Instead, they are almost equally important for STS. There are two pieces of evidence:
>
> First, the Spatial Ensemble (SE) by itself is capable to achieve comparable performance as TMA. Table 4 in the manuscript shows that when using neuron-wise (NW) granularity, SE achieves 71.6\% top-1 accuracy, which is only -0.2\% lower than BYOL with TMA. In Table 1, the performance gap between SE and TMA appears relatively large because Table 1 adopts layer-wise (LW) instead of the neuron-wise granularity.
>
> Second, if we use SE as the baseline and adding TMA to implement STS, it appears that adding TMA only brings incremental improvement over the SE baseline. In another word, if we abandon the prior that TMA is earlier than SE, these two model smoothing mechanisms do look dual to each other and are almost equally important for STS.
>
> > 2. Why does this work choose LW as the default method rather than CW?
>
> Thanks for your careful review. We agree that STS (CW) can be employed as a competent alternative. Generally, STS (LW) and STS (CW) are very close to each other, w.r.t. both the speed and accuracy. Table 4 in the manuscript shows that STS~(CW) is 0.05\% slower and 0.1\% more accurate than STS (LW). In this paper, we use STS (LW) more frequently because we cared more about the (experimental) efficiency. That being said, STS (CW) is a competent alternative if someone cares more about accuracy.
>
> > 3. The improvements of STS on ImageNet are incremental in most methods.
>
> The improvements on ImageNet may seem incremental but are still valuable. Based on multiple state-of-the-art methods, the improvements are general and consistent. Moreover, under data corruption, the advantage of STS is even larger. On Imagenet-C, STS yields +1.5\% top-1 accuracy improvement under the combination of 75 corruptions. Besides, considering a single corruption from these 75 combinations, STS leads to an improvement of around 0.5\% to 5.6\%.

---

### Official Review · Reviewer_P4s6 · 2021-07-16

**Rating:** 8
**Confidence:** 4

**Summary:**

The authors propose a new framework for constructing teacher models in student-teacher model training, which involves a combination of spatial smoothing (piecewise copying of students to teacher) and the usual temporal smoothing (trajectory-wise parameter averaging).   They situate spatial smoothing inside a more general framework of spatial-temporal smoothing, and recover temporal smoothing and the newly proposed spatial smoothing as special cases of spatial-temporal smoothing.  They go on to show its utility in two learning scenarios (semi-supervised learning and self-supervised learning).

**Ethical Concerns:**

No ethical concerns are apparent to me

**Limitations And Societal Impact:**

The authors do not discuss the limitations of their work directly, but do indirectly present evidence that shows the limitation of how much benefit STS offers in different scenarios.

I would like to see a statistical analysis of the benefits of STS (see above)

**Main Review:**

This paper presents a new model-smoothing technique for student-teacher training.  The paper is written with exceptional clarity.  It was always clear from first reading what the authors were attempting, what motivated them, and what they observed. Though the mechanism of spatial smoothing is under-explored, they present compelling evidence that this smoothing method for student-teacher training offers benefits over existing smoothing.  For me, it's a clear accept.

Notes:

- In section 4.3 the authors experiment with the granularity of the spatial smoothing.  They investigate layer-wise versus channel-wise, and neuron-wise spatial transfer.  In 234-247, the authors report: "As shown in the top of Table 4, SE using NW outperforms its counterpart using LW and CW and achieves comparable performance with TMA. The result reveals that SE itself benefits from the fine-grained spatial ensemble."   I feel as if there is more the authors could say here.  It feels as if this experiment could benefit from the perspective of a model averaging argument, or from analyzing the correlation between model components that NW versus CW entail.  I encourage the authors to consider following up on this.

- For Table 2, the results are so similar that it's unclear if there is any benefit of STS smoothing over TMA.  IF possible, could this experiment be replicated several times and the distributions reported?

- The caption for Figure 3 should indicate which panel represents model parameters, and which presents generated supervision signals.  I can guess that they are in order induced by the caption, but more clarity here would be appreciated.



**Time Spent Reviewing:**

5

---

> ### Author Response · Authors · 2021-08-10
> **Thanks and response to concerns**
>
> We thank the reviewer for the positive and constructive comments.
>
> > 1. More discussions about the granularity of SE from the perspective of a model averaging argument, or from analyzing the correlation between model components that NW versus CW entail.
>
> Thanks for your suggestion. We agree that the impact of the SE granularity deserves deeper discussion. In our view, fine-grained spatial ensemble NW is more flexible since CW and LW can be regarded as two special cases of NW. Besides, NW makes the teacher update more smoothly and fluently, which may be the possible reason why SE with NW performs better than its counterpart using LW and CW. We will add more discussions in our revised version and conduct more in-depth investigations on this issue in the future.
>
> > 2. Could the experiments in Table 2 be replicated several times and the distributions reported?
>
> We repeat the experiments by five times following your suggestion. In overall, STS brings consistent and general improvements over TMA. For MoCo, TMA obtains top-1 accuracy of **67.42 $\pm$ 0.07\%** (67.5\%, 67.3\%, 67.4\%, 67.4\%, and 67.5\%), while STS achieves **67.76 $\pm$ 0.05\%** (67.8\%, 67.7\%, 67.8\%, 67.7\%, and 67.8\%).  For BYOL, TMA obtains top-1 accuracy of **71.76 $\pm$ 0.10\%** (71.8\%, 71.8\%, 71.7\%, 71.9\%, and 71.6\%), while STS achieves **72.78 $\pm$ 0.07\%** (72.7\%, 72.8\%, 72.7\%, 72.8\%, and 72.9\%). We will release our codes to the community for the convenience of reproducity if the paper is accepted.
>
> > 3. The caption for Figure 3 should indicate which panel represents model parameters, and which presents generated supervision signals.
>
> Thanks. We will make this part more clear by modifying the first sentence of the caption to ''Illustration of mean square error of teacher parameters (Left) and generated supervision signals (Right) at two adjacent epochs."

---

> > ### Comment · Reviewer_P4s6 · 2021-09-10
> > **Thanks!**
> >
> > I'm glad that the small changes I've suggested have been addressed by the authors.

---

### Official Review · Reviewer_um8Q · 2021-07-17

**Rating:** 6
**Confidence:** 3

**Summary:**

This paper proposes a novel technique to perform model smoothing in the context of the student-teacher training framework for tasks like self-supervised learning and semi-supervised learning. In essence, the proposed method involves randomly sampling a small portion of the student model and use the selected portions to replace its teacher counterpart. The authors of the paper demonstrate that the proposed method is also complementary to the commonly used temporal moving average method and when combined together, can yield state-of-the-art performance in multiple benchmark datasets.

**Limitations And Societal Impact:**

The authors adequately addressed the limitations and potential negative societal impact of their work.

**Main Review:**

All in all, the paper is very well written and easy to follow. Although the proposed method can be quite simple, it can be of great significance to machine learning practitioners as a straightforward and convenient way to improve performance. Moreover, the authors of the paper conduct a series of experiments to demonstrate the effectiveness of the paper. All in all, while I think the novelty can be somewhat limiting because the idea is quite a simple one, this paper is a good submission in terms of the quality of the work/paper. As such, I will be a little bit conservative with my score for now and would like to hear the opinions of other reviewers. I would be happy to raise my score to a 7 if there is a consensus among the reviewers.

Questions:

1. It is stated in the paper that a crucial factor for a teacher model to yield good performance lies in its stability/small variance, and the proposed spatial ensemble is a way to reduce variance. However, intuitively, doing so (especially when combined with TMA) can potentially harm the "accuracy" of the teacher model? Is this not an important aspect for teacher models to be effective?

**Time Spent Reviewing:**

1 hour

---

> ### Author Response · Authors · 2021-08-10
> **Thanks and response to concerns**
>
> We thank the reviewer for the positive and constructive comments.
>
> > 1. Reducing variance can potentially harm the accuracy of the teacher model.
>
> We guess your concern is that the small variance of the teacher model may slow down the speed of model update/convergence, and thus may potentially harm the accuracy. If so, we hope to resolve your concern by two observations:
>
> First, model smoothing is necessary and promotes convergence of the student-teacher framework, as illustrated in Section 3.1 (lines 130-140). Without model smoothing, the teacher model and its output signals will undergo substantial instability, leading to the failure of convergence.
>
> Second, a combination of SE and TMA with proper smoothing intensity benefits the student-teacher learning performance. Figure 4 in the submitted manuscript demonstrates the performance varying with different smoothing intensities. As is shown, too weak or too strong model smoothing both leads to sub-optimal performance. A good choice is to use a larger value of momentum and a smaller one of preserving probability, or vice versa, as discussed in lines 220-226.
>
> In general, the benefits of model smoothing outweigh the potential disadvantages~(relatively slow update). We can take full advantage of model smoothing by properly adjusting its intensity.

---

### Decision · Program_Chairs · 2021-09-27

**Decision:**

Accept (Poster)

**Comment:**

There is a consensus among the reviewers that this paper presents significant results and is of interest to the conference.